# A Surface-Mount Substrate-Integrated Waveguide Bandpass Filter Based on MEMS Process and PCB Artwork for Robotic Radar Applications

**DOI:** 10.3390/mi17010072

**Published:** 2026-01-02

**Authors:** Yan Ding, Jian Ding, Zhe Yang, Xing Fan, Wenyu Chen

**Affiliations:** 1School of Information and Control Engineering, North China Institute of Science and Technology, Langfang 101601, China; fanguan123@126.com (X.F.); 17526712912@163.com (W.C.); 2Key Laboratory of Brain-Computer Interface Technology Application of the Ministry of Emergency Management, Langfang 101601, China; 3National Astronomical Observatories of the Chinese Academy of Sciences (NAOC), Beijing 100101, China; dingjian@bao.ac.cn; 4Nanjing Electronic Devices Institute, Nanjing 210016, China; zheyang_nedi@163.com

**Keywords:** bandpass filter, slot-line resonators, substrate-integrated waveguide (SIW), out-of-band rejection

## Abstract

To address the pressing need for compact and highly reliable perception systems in autonomous mobile robots, a compact bandpass filter (BPF) integrating slot-line resonator with substrate-integrated waveguide (SIW) technology for robotic millimeter-wave radar front ends was proposed. By integrating slot-line resonators between adjacent SIW cavities, the proposed design effectively increases the filtering order without increasing the layout area. This approach not only generates extra transmission poles but also creates a sharp transmission zero at the upper stopband, thereby significantly enhancing out-of-band rejection. This characteristic is crucial for robotic radar operating in complex and dynamic environments, as it effectively suppresses out-of-band interference and improves the system signal-to-noise ratio and detection reliability. To validate the performance, a prototype filter operating in the 24.25–27.5 GHz passband was fabricated. The measured results show good agreement with simulations, demonstrating low insertion loss, compact size, and wide stopband. Finally, to validate its compatibility with robotic radar modules, the chip was assembled onto a PCB using surface-mount technology. The responses of the bare die and the packaged module were then compared to evaluate the impact of integration on the overall RF performance. The proposed design offers a key filtering solution for next-generation high-performance, miniaturized robotic perception platforms.

## 1. Introduction

Substrate-integrated waveguide technology has garnered significant research interest in millimeter-wave systems, particularly for millimeter-wave radar front ends that demand stringent requirements on size, cost, and performance, owing to its compelling advantages of high-quality factor, minimal insertion loss, low cost, and fabrication simplicity [1,2,3]. In applications such as autonomous mobile robots, radar sensors must achieve highly reliable perception within extremely compact spaces. This imposes critical demands on core filtering components, including low loss, high selectivity, and extreme miniaturization.

Despite their advantages, standard SIW resonators exhibit larger physical dimensions compared to alternative planar guided-wave technologies. Recent miniaturization efforts have therefore focused on size-reduced SIW cavity filter implementations. A variety of techniques have been proposed to miniaturize SIW bandpass filters. The primary miniaturization approach employs planar etching-based technology, including half-mode [4], quarter-mode [5], eighth-mode [6], and sixteenth-mode [7] SIW cavities. However, they suffer from extra radiation loss from the open boundary. The second miniaturization approach exploits vertical dimension reduction, with the ridged substrate-integrated waveguide (RSIW) [8] being the most representative implementation. It enables size reduction without compromising radiation performance, albeit with increased fabrication complexity, specifically requiring layer alignment tolerance improved from ±8 μm to ±3 μm. In addition, loading-based miniaturization integrates structures like electromagnetic band-gap elements [9], defected ground structures [10], or complementary split-ring resonators (CSRRs) integrated on the SIW surfaces [11]. These modifications alter electromagnetic field distributions to reduce footprint. However, the hybrid configurations compromise SIW cavity integrity and electromagnetic shielding, inducing radiation loss and significant $Q_{u}$ degradation.

Simultaneously, to cope with strong interference in complex and dynamic environments, filters for robot radar must also possess superior higher-order mode suppression and wide stopband characteristics. Existing methods [12,13,14,15,16,17], such as optimizing feed port positions [12,13], introducing cross-coupling [14], or loading perturbing vias [15], improve performance but often at the expense of structural complexity or size.

In response to these challenges, a novel fourth-order SIW bandpass filter suitable for compact robot radar front ends is proposed. The design innovatively etches slot lines between adjacent SIW cavities, allowing them to function simultaneously as resonant elements and coupling structures. This approach not only generates additional transmission poles without occupying extra layout area but also naturally creates a transmission zero in the upper stopband, thereby achieving enhanced out-of-band rejection and wide stopband response. This is crucial for improving the anti-interference capability of robot radar in spectrally dense environments. The adopted single-layer linear topology is fully compatible with standard silicon-based MEMS processes, ensuring excellent manufacturability and consistency, and facilitating integration with the radio-frequency modules of robot radar systems.

## 2. Design and Simulation

### 2.1. Filter Configuration

The configuration of the proposed four cavity SIW filter is shown in Figure 1a. Four resonant cavities are separated by three pairs of symmetric through-holes, with slot-line resonators arranged between each pair of through-holes. The open-ended slot lines are etched on the surface and can be equivalent to a shorted half-wavelength uniform impedance resonator. The boundary via array physically confines the electromagnetic fields to form the SIW structure. Coplanar waveguide (CPW) excitation is employed at the input/output ports. Figure 1c shows its coupling scheme, where the resonant $R_{i}$ (i = 1, 3, 5, 7) values represent the SIW cavities; $R_{i}$ (i = 2, 4, 6) values represent the slot-line resonators. Electric coupling governs the interaction between the SIW cavity and the slot line, while magnetic coupling mediates energy exchange between neighboring resonant cavities.

All the resonant frequencies of ${TE}_{m0n}$ modes can be expressed by using the method in [14]:(1)$$f_{{TE}_{m0n}}=\frac{c_{0}}{2\sqrt{\varepsilon_{r}\mu_{r}}}\sqrt{{\left(\frac{m}{w_{eff}}\right)}^{2}{+(\frac{n}{l_{eff}})}^{2}}.$$

In this expression, m and n correspond to the resonant mode indices along the transverse directions x and y, while $l_{eff}$ and $w_{eff}$ account for the effective electrical length and width of the SIW cavity. The constants $\varepsilon_{r}$ and $\mu_{r}$ denote the relative permittivity and permeability of the dielectric substrate, and $c_{0}$ is the speed of light in vacuum.

The slot-line resonator between adjacent cavities operates as a half-wavelength resonant structure, providing the dominant electric coupling between adjacent SIW cavities. Its physical length governs the coupling phase and is the primary parameter for locating the transmission zero. Figure 1c shows the equivalent circuit mode of the proposed filter, which elucidates the underlying coupling mechanism. Each SIW cavity is represented by a parallel LC resonator (represented by$\;L_{c}$ − $C_{c}$)). The slot-line resonator is modeled as a grounded parallel LC resonator (represented by $\;L_{s}$ − $C_{s}$), and a via structure characterized by parallel inductance $L_{via}$ [16]. At the slot-line resonator’s resonant frequency, its low impedance effectively shorts the coupling path to the ground, thereby blocking signal transmission between adjacent cavity resonators and producing a transmission zero in the stopband.

To facilitate the design of the proposed filter, full-wave electromagnetic (EM) simulations were conducted using Ansys HFSS on a simplified two-cavity prototype. The external quality factor $Q_{e}$ is primarily governed by the CPW feed geometry, specifically the feed window width $W_{s}$ and the signal-to-ground gap $W_{g}$. Meanwhile, the inter-cavity coupling strengths are engineered through the dimensions of the slot-line features. The relationship between $Q_{e}$ and $W_{s}$ for different values of $W_{g}$ is shown in Figure 2a. The weak cross-coupling path between the outer cavities, denoted as $K_{13}$, is essential for generating a transmission zero. As confirmed by eigenmode analysis in Figure 2c, $K_{13}$ remains below 0.04 across the design space, which is significantly weaker than the main couplings $K_{12}$ and $K_{23}$ shown in Figure 2b. However, $K_{13}$ can be effectively controllable by adjusting the coupling window size and slot-line length $l_{4}$. The parametric studies in Figure 2 are conducted to capture the fundamental trends for initial design guidance. The parameters extracted from these analyses serve as the critical starting point for synthesizing the four-cavity layout, whose final dimensions are then determined through a holistic full-wave electromagnetic optimization to precisely account for multi-cavity interactions [18,19,20].

### 2.2. Filter Design, Tuning, and Analysis

A four-cavity filter without slot-line resonators is initially analyzed. Signal transmission between resonant cavities primarily occurs through inter-cavity coupling windows, where the size of coupling window $c_{1}$ governs signal integrity. Variations in the $c_{1}$ induce significant alterations in resonator responses, manifesting as modified attenuation levels or reduced signal loss across cavities. Balanced pole distribution within the passband can be achieved by optimizing $\;c_{1}$, as shown in Figure 3a. Figure 3b illustrates the impact of the CPW feed window on the return loss. By adjusting the parameter $W_{s}$, the pole position of the first resonant cavity can be effectively altered, which also influences the second resonant cavity. This effect is due to changes in the TE_101_ mode electric field distribution within the first cavity caused by variations in $W_{s}$. These changes subsequently affect the second pole, ultimately manifesting in the overall scattering response of the filter.

To illustrate the mechanism by which higher-order modes are suppressed in the proposed structure, simulations are conducted for two filter configurations: one without the slot-line resonator and one incorporating it, as shown in Figure 4. It can be seen that the introduction of the slot line creates additional passband poles and upper stopband zero, improving the out-of-band rejection at 30 GHz from −18 dB to −56 dB. For the proposed four cavities filter, the slot-line resonators provide a coupling path for the adjacent SIW resonators. To further evaluate the impact of the slot-line resonators on the in-band phase response while improving the stopband, the phase response and group delay within the passband is critically examined. Figure 4b compares the passband phase response of the filter with and without slot-line resonators. Both phase curves maintain high linearity, confirming that the incorporation of slot-line resonators does not compromise the phase linearity within the passband. A more quantitative assessment is provided by the group-delay analysis, as shown in Figure 4c. Notably, the filter with slot-line resonator exhibits a reduced peak-to-peak group delay variation of approximately 34 ps, compared to 78 ps for the baseline structure. This improvement in group-delay flatness indicates a lower level of phase nonlinearity, which is beneficial for preserving signal integrity.

In order to achieve a larger bandwidth, the length $l_{3}$ is optimized, as shown in Figure 5. When the $l_{3}$ is tuned, not only the passband width is influenced but also the transmission zero shifts. Through tuning, a 24.25–27.5 GHz passband is realized, with the suppression exceeding 40 dB at 20 GHz and 30 GHz. The insertion loss within the passband remains below 0.94 dB.

Figure 6 shows the electric filed distribution in the passband 24 GHz and in the upper stop band 32 GHz. The distinct patterns provide physical insight into the operating and rejection mechanisms. Within the passband at 24 GHz, the electric field is strongly concentrated along the slot-line resonators, with maximum intensity occurring at their open ends. This confirms that the slot-line resonators serve as the primary electric coupling path between adjacent SIW cavities, directly facilitating the formation of the three in-band transmission poles. In contrast, the field within the SIW cavities themselves remains relatively weak, indicating that energy is efficiently transferred via the slot-line resonators rather than being confined to individual cavities. Within the stopband at 32 GHz, the field distribution exhibits a significant change in which the field along the slot-line resonators diminishes substantially, while strong, localized standing-wave patterns emerge within each SIW cavity. This shift indicates that the slot-line resonators cease to be an efficient coupling path near their own resonant frequency. Instead, energy becomes trapped within individual cavities, effectively decoupling them and giving rise to the observed upper stopband transmission zero. Therefore, the contrasting field distributions visually corroborate the dual role of the slot-line resonators, i.e., as efficient coupling elements in the passband and as sources of a transmission zero in the stopband through path decoupling, which is the key to achieving wide stopband suppression. The final geometric parameters of the 24.25–27.5 GHz SIW bandpass filter are summarized in Table 1.

## 3. Results and Discussion

### 3.1. BPF Fabrication and Measurement

The SIW filter is realized on a 400 μm thick silicon wafer through standard microelectromechanical system (MEMS) fabrication techniques. A sequence of lithography, etching, and electroplating processes were used in the fabrication, as shown in Figure 7. A high-resistance silicon is deposited and patterned over the substrate (a). Through-holes are etched by etching (b). Sputtering and patterns on the surface are formed (c). Electroplating is performed on the metal to make it thicker (d). The glue is removed and the thickness of the metal is controlled by IBE (e). Sputtering and electroplating are performed on the back of the wafer (f).

The fabricated chip is visualized in Figure 8a, demonstrating the successful structure. Figure 8b presents the simulated and measured frequency responses, revealing a center frequency of 25.875 GHz and a 2-dB bandwidth of 3.25 GHz. In the passband, three distinct resonant poles are clearly observed, while a transmission zero appears in the stopband, confirming the desired filtering behavior. The measured in-band return loss exceeds 14 dB, and the minimum insertion loss is 1.23 dB, which is slightly higher than the simulated value of 0.94 dB. This discrepancy is likely due to fabrication-induced deviations, including non-uniformities in deep silicon etching, variations in via hole diameters, slot-line widths, and other process-related tolerances.

To evaluate the performance advantages of the proposed design, a comparative analysis with recently reported bandpass filters is provided in Table 2. While PCB-based SIW filters [2] offer low insertion loss, they suffer from large footprints. Filters employing controllable electric and magnetic mixed coupling [18] exhibit relatively high insertion loss, limited out-of-band suppression, and larger area. In contrast, the present filter achieves a broader frequency response, lower insertion loss, and superior out-of-band rejection, demonstrating its effectiveness for compact high-performance applications. Compared with the filter reported in [19], the proposed design offers a broader bandwidth, lower loss, and improved the out-of-band rejection. The hybrid SIW-CSRR filter in [20] demonstrates effective stopband extension in the X-band with a compact layout, albeit with a narrow bandwidth and no transmission zeros. Another hybrid SIW filter in [21] successfully introduces independently tunable transmission zeros on both sides of the passband, enabling high selectivity and design flexibility. However, this approach requires a more complex coupling topology and results in a significantly larger footprint. By contrast, the proposed filter realizes comparable selectivity and even deeper stopband suppression through a simplified electric-coupling scheme, while occupying a significantly smaller footprint, making it more suitable for compact millimeter-wave front ends.

To further support the practical viability of the employed process, it is noted that a filter from the same fabrication family, using an identical substrate, metallization materials, and process steps, underwent high-temperature storage (HTS) testing at 150 °C for 500 h. The post-stress RF characterization showed no measurable performance degradation, thereby confirming the thermal stability and reliability of the shared platform.

### 3.2. PCB Integration

To facilitate integration into robotic radar front-end systems, the PCB implementation including layout, flip-chip assembly, and RF characterization is discussed. Figure 9a presents a three-dimensional rendering of the assembled sub-module, which integrates the fabricated SIW bandpass filter chip onto a dedicated interposer board. The PCB stack up employs Panasonic M6 laminates ($\varepsilon_{r}$ = 3.6), with a substrate thickness of 0.254 mm, selected to ensure impedance compatibility and low propagation loss at millimeter-wave frequencies. To accommodate the filter proposed in this letter, the PCB needs to exhibit low loss at the corresponding frequency band. The PCB signal line adopts a tapered structure, consisting of three sections: the RF input section, the impedance transformation section in the middle, and the section connected to the chip feedline. Conductive vias with a radius of 0.25 mm are placed 0.82 mm away from the signal lines. Four alignment holes are placed at the corners of the PCB to accommodate mounting screws and provide mechanical stability during measurement of the device under test. The SIW bandpass filter chip is integrated onto the board via flip-chip bonding. To evaluate the intrinsic transmission loss of the PCB itself, a through-path structure consisting of two matched feedlines was implemented, as illustrated in Figure 9b. This calibration path exhibits a return loss exceeding 14.8 dB and an insertion loss below 2.2 dB across the operating band. The residual loss is primarily attributed to imperfections in the calibration procedure, connector repeatability, and minor misalignments in the bonding interface.

Figure 10 shows the comparison of the BPF and surface-mounted filter on the PCB. The errors introduced by the connector, calibration, and soldering have been de-embedded. The insertion loss of the filter increases by less than 0.7 dB after surface-mounting onto the PCB, and the out-of-band suppression is slightly affected. The measured 0.7 dB increase in insertion loss after PCB integration is attributed to the chip-to-board transition, primarily arising from the flip-chip interconnect resistance and a minor impedance mismatch at the interface. This incremental loss is inherent to the physical assembly process and does not compromise the filter’s core functionality. The out-of-band suppression remains robust, confirming the practical suitability of the design. The analysis presented in this section provides valuable guidance for the real-world implementation of the filter chip.

## 4. Conclusions

This paper presents a wideband and compact substrate-integrated waveguide BPF based on the MEMS process. The measured results indicate that the proposed filter has a low insertion loss, compact size, and high out-of-band rejection. These characteristics confirm its excellent suitability and promising potential for integration into front-end modules of compact robotic millimeter-wave radar systems.

## Figures and Tables

**Figure 1 micromachines-17-00072-f001:**
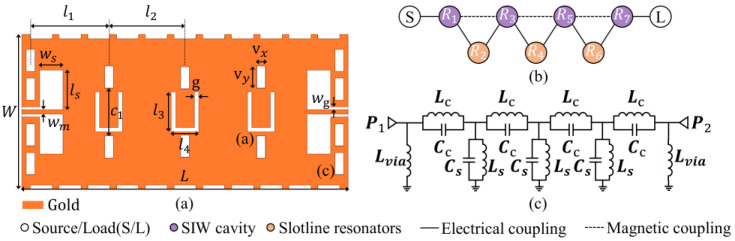
Layout of the proposed SIW filter. (**a**) Top view with geometric parameters. (**b**) Magnified slot-line. (**c**) Equivalent circuit model.

**Figure 2 micromachines-17-00072-f002:**
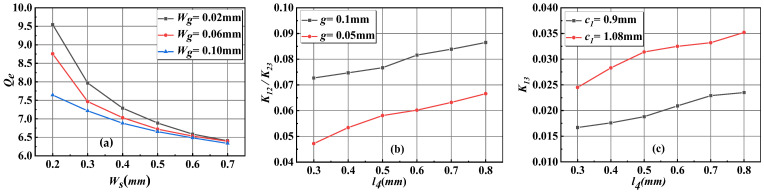
(**a**) Variation in external quality factor $Q_{e}$ with feed window width $W_{s}$ and signal-ground gap $W_{g}$; (**b**) coupling coefficients$\;K_{12}/K_{23}$ versus slot-line width g and length $l_{4}$ with $c_{1}$ fixed at 0.9 mm; (**c**) influence of coupling window size $c_{1}$ and $l_{4}$ with g fixed at 0.1 mm on coupling coefficients $K_{13}$.

**Figure 3 micromachines-17-00072-f003:**
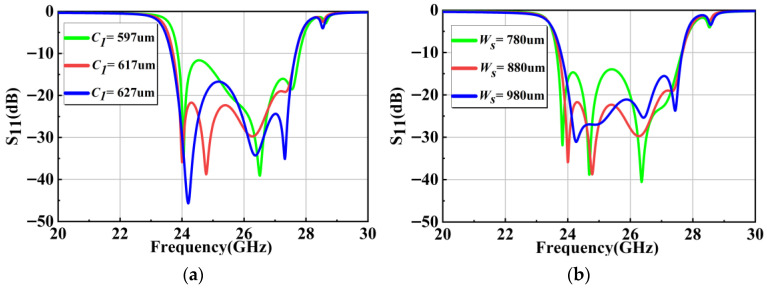
Simulated frequency response of the four-cavity filter without slot lines as a function of coupling window size $c_{1}$ (**a**) and feed window width $W_{s}$ (**b**).

**Figure 4 micromachines-17-00072-f004:**
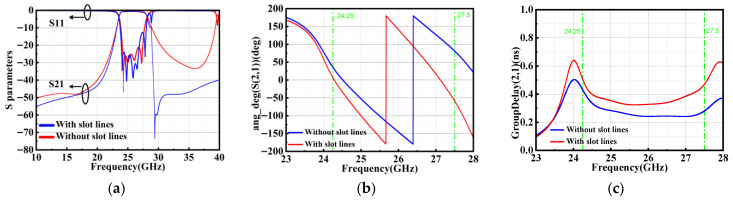
Comparison of the simulation of SIW filter coupling without slot-lines and with slot-line resonators: (**a**) S-parameters, (**b**) phase response, and (**c**) group delay.

**Figure 5 micromachines-17-00072-f005:**
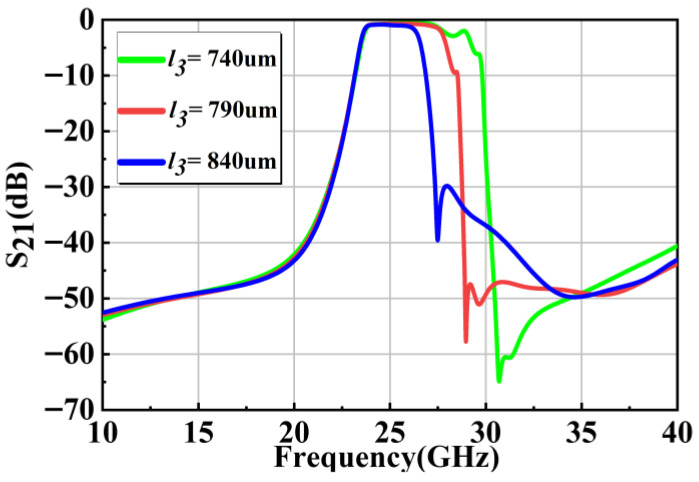
Four-cavity filter model with the slot-lines, and S_21_ versus $l_{3}$.

**Figure 6 micromachines-17-00072-f006:**
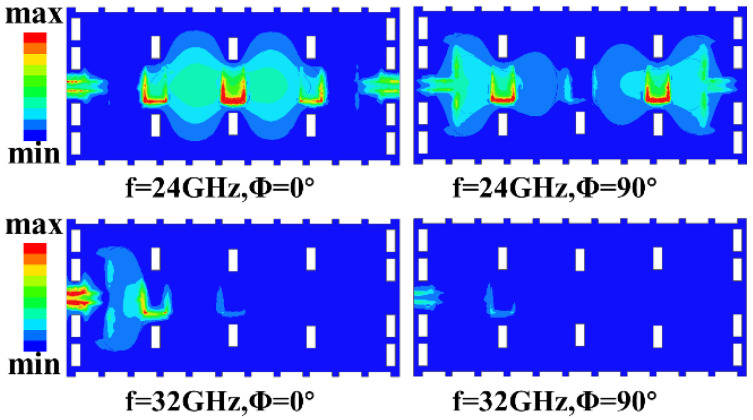
Electric field distribution at 24 GHz and 32 GHz.

**Figure 7 micromachines-17-00072-f007:**
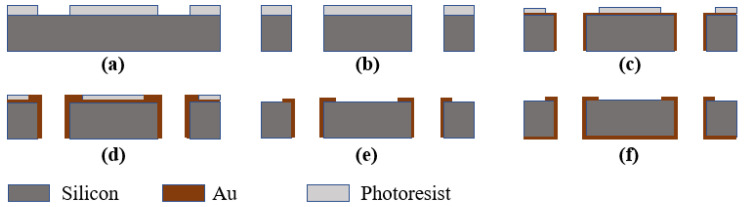
MEMS-based fabrication steps of the SIW filter: (**a**) lithography, (**b**) etching, (**c**) sputtering, patterned, (**d**) electroplating, (**e**) IBE, and (**f**) sputtering and electroplating on the back.

**Figure 8 micromachines-17-00072-f008:**
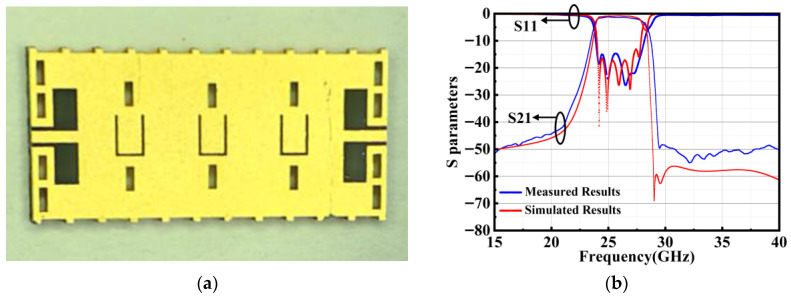
(**a**) Micrograph of the chip; (**b**) simulated, measured results for proposed SIW filter.

**Figure 9 micromachines-17-00072-f009:**
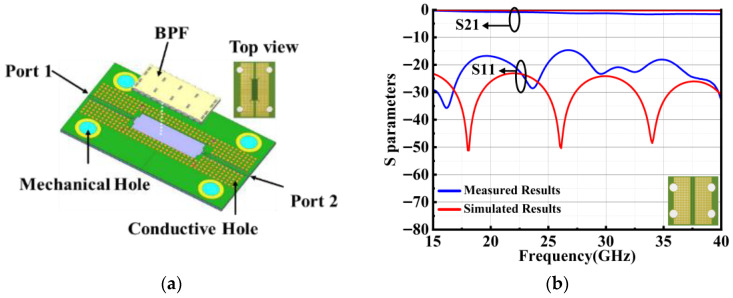
(**a**) 3D view of the PCB with the mounted SIW BPF chip. (**b**) Simulated and measured S-parameters of the PCB.

**Figure 10 micromachines-17-00072-f010:**
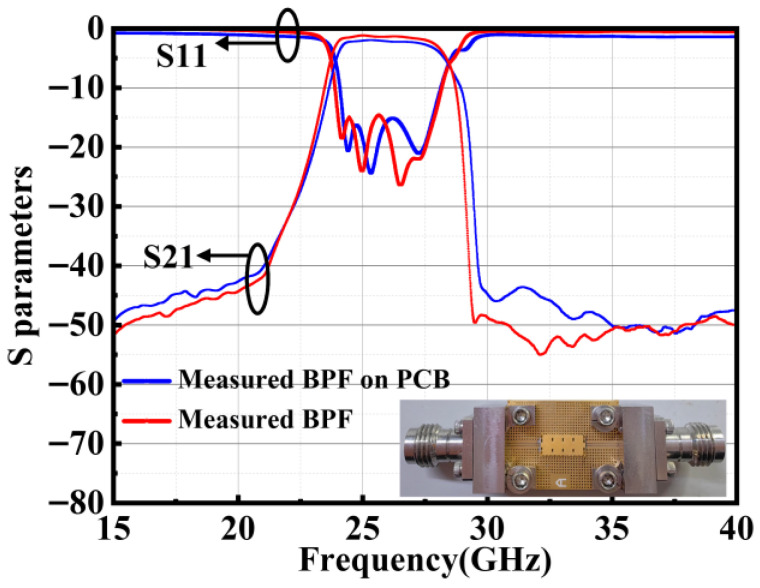
Comparison of BPF and surface-mounted filter on PCB.

**Table 1 micromachines-17-00072-t001:** Geometric parameters of the 24.25–27.5 GHz SIW BPF after optimization (unit: μm).

Symbol	Value/μm	Symbol	Value/μm
L	7400	$w_{s}$	535
W	3540	$l_{s}$	900
$l_{1}$	1775	$c_{1}$	1234
$l_{2}$	1725	g	75
$w_{g}$	66	$l_{3}$	790
$w_{m}$	200	$l_{4}$	615

**Table 2 micromachines-17-00072-t002:** Performance comparison of the proposed SIW BPF with state-of-the-art designs.

Ref.	Form	$f_{0}$ (GHz)	FBW (%)	IL (dB)	S21 Suppression	Size (mm^3^)
Prop.	SIW	25.875	12.5	1.4	$-50\;\mathrm{dB}@1.55f_{0}$	10
[2]	SIW	20.2	10	0.9	$-40\;\mathrm{dB}@1.5f_{0}$	197
[21]	SIW	27	7.4	2	$-20\;\mathrm{dB}@1.26f_{0}$	127
[22]	SIW	22	4.5	3.57	$-22\;\mathrm{dB}@1.68f_{0}$	25.5
[23]	hybrid	10	12	1.59	$-20\;\mathrm{dB}@f_{0}$	107.5
[24]	hybrid	11.51	6.78	1.42	$-28\;\mathrm{dB}@1.24f_{0}$	174

## Data Availability

Data will be made available on request.

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
