# Peer review of "A Surface-Mount Substrate-Integrated Waveguide Bandpass Filter Based on MEMS Process and PCB Artwork for Robotic Radar Applications"

_micromachines, 2026, doi:10.3390/mi17010072_

Round 1

Reviewer 1 Report

Comments and Suggestions for Authors

The paper proposes an SIW-based bandpass filter. By loading SRs, it introduces one passband pole and one upper-stopband transmission zero. It verifies the influence of various parameters on the filter performance, and finally provides a chip-to-PCB integration validation. The application need is clear, the structural idea is simple and effective, and the PCB–chip verification forms a complete engineering loop. However, the mechanism explanation is insufficient, the control variables in the parameter extraction are not well handled, and there is no transmission zero on the lower side of the band.

  1. From the beginning of the manuscript up to Fig. 2, the relevant design parameters are not clearly stated. In Fig. 2b, the authors present the relationship between the adjacent coupling coefficients and the slot length under different U-shaped slot widths; however, what is the window spacing (c1) used for this extraction? In Fig. 2c, the authors show (K13) versus (l4) under different (c1) values and state that (K13) is much smaller than (K12) and (K23) in Fig. 2b, but the control variables are not clearly defined, especially the effect of (g). Present the influence of (g) and (c1) on (Kij) in a more intuitive way, or provide results under a consistent set of fixed variables so that the comparison of (Kij) is made with proper control of parameters?
  2. There have been many reports showing that introducing U-shaped slots can generate a high-frequency radiation/transmission zero. I noticed that the responses for (l3 = 790um and 810um) appear almost identical. The authors can consider adjusting the (l3) sweep interval (e.g., 760/790/820 um or 775/790/805 um) to more clearly demonstrate the expected trend that the radiation/transmission-zero frequency decreases as (l3) increases?
  3. Please add more explanatory discussion for the field distributions in Fig. 6? Stating only that they help suppress spurious bands seems a bit too brief. It would be helpful to enrich this part with one or two mechanism-oriented explanations, for example: clarifying the dominant energy/field distribution differences between the SRs and the SIW cavities in the passband versus the stopband; linking the field behavior to the equivalent circuit or coupling paths to explain why the observed field distributions correspond to the additional pole and the upper-stopband zero; and, if possible, providing a brief comparison of field intensity on key cross-sections or an energy/field-intensity ratio.
  4. The authors mention that the SR can be modeled as a half-wavelength uniform-impedance resonator and discuss its equivalent electrical length and admittance, but did not provide a corresponding parameter-extraction method for the equivalent circuit, such as the relationship between the capacitance value and the geometric dimensions.
  5. The authors state that the two-cavity model may deviate when extended to the four-cavity case. How large is this deviation, and could it be significant? Is there any related literature addressing this issue? I suggest adding relevant references and clarifying the potential impact on the final design.
  6. In Table 1, (c1 = 1234um), which is not within the (c1 = 0.9–1.08mm) range used in Fig. 2. Would this introduce some inconsistency or error in the parameter extraction and comparison?
  7. The comparison references in Table 2 are not new. At least some should be within the last five years; otherwise, it is difficult to demonstrate the superiority of the present work.  
  8. The unit system in Fig. 2 and the related text seems mixed. Please consider unifying the units to either (um) or (mm).
  9. Could the frequency span in Fig. 3 be narrowed so that the resonant features are displayed more clearly, for example, 20–35 GHz? Alternatively, adding an additional zoomed-in plot over 20–35 GHz would better illustrate the in-band poles and the tuning trend.
  10. Regarding the mechanism of the transmission zero, this work is different from approaches based on mode selection or non-resonating nodes; here it mainly comes from the electric coupling path between the slotline SR and the SIW cavity. As for the transmission pole, few papers explicitly emphasize this concept, or the concept itself may not be particularly critical in this context. The fabrication and integration part is relatively well completed. The chip-on-carrier/PCB implementation and the module-level validation provide valuable guidance for practical filter applications, which is a major highlight. (1)“A Compact, Hybrid SIW Filter With Controllable Transmission Zeros and High Selectivity” is a typical example with controllable dual-side transmission/radiation zeros and can be used as a key reference for comparison. (2)“Broadband external rejection miniaturized SIW filter based on C-shaped symmetric structure”: the equivalent-circuit diagrams in Fig. 2 and Fig. 5 of that paper can be referenced, and a similar representation could be added here. (3)“Compact Hybrid Bandpass Filter Using SIW and CSRRs with Wide Stopband Rejection” (or similar works) could also be included in the comparison.

In summary, the novelty of this paper appears limited, but it does present a relatively new structural idea. The PCB–chip engineering loop is a strong highlight. The technical comments above can be used as references for revision. However, adding necessary theory or mechanism explanations at appropriate places is still important. Overall, the manuscript is generally aligned with the scope of Micromachines. With appropriate revisions should be able to meet the acceptance requirements.

Reviewer 2 Report

Comments and Suggestions for Authors
  1. The paper mentions that the introduction of slot-line resonators (SRs) enhances out-of-band rejection, but it does not clearly address their impact on in-band insertion loss and group delay. Could you further analyze whether the use of SRs, while improving stopband suppression, adversely affects in-band phase response or signal integrity?

  2. In the PCB integration section, the authors note an approximate 0.7 dB increase in insertion loss after surface mounting, but do not thoroughly investigate the specific sources of this additional loss (e.g., solder joint impedance, impedance mismatch, radiation loss). Could you provide further analysis or experimental data to clarify the loss mechanisms and suggest possible optimization approaches?

  3. Table 2 compares the proposed filter with prior works in terms of size and suppression, but does not address important practical metrics for radar applications such as power handling capability or thermal stability. Has the filter's performance been validated under high-temperature, high-power, or varying temperature conditions? Are there any reliability test data available?

Comments on the Quality of English Language

English is good

Round 2

Reviewer 1 Report

Comments and Suggestions for Authors

The authors have satisfactorily addressed all of my comments. I recommend acceptance of the manuscript in its present form.